

# Building a Korean morphological analyzer using two Korean BERT models

Yong-Seok Choi,  Yo-Han Park and  Kong Joo Lee

Department of Radio and Information Communications Engineering, Chungnam National University, Daejeon, South Korea

## ABSTRACT

A morphological analyzer plays an essential role in identifying functional suffixes of Korean words. The analyzer input and output differ from each other in their length and strings, which can be dealt with by an encoder-decoder architecture. We adopt a Transformer architecture, which is an encoder-decoder architecture with self-attention rather than a recurrent connection, to implement a Korean morphological analyzer. Bidirectional Encoder Representations from Transformers (BERT) is one of the most popular pretrained representation models; it can present an encoded sequence of input words, considering contextual information. We initialize both the Transformer encoder and decoder with two types of Korean BERT, one of which is pretrained with a raw corpus, and the other is pretrained with a morphologically analyzed dataset. Therefore, implementing a Korean morphological analyzer based on Transformer is a fine-tuning process with a relatively small corpus. A series of experiments proved that parameter initialization using pretrained models can alleviate the chronic problem of a lack of training data and reduce the time required for training. In addition, we can determine the number of layers required for the encoder and decoder to optimize the performance of a Korean morphological analyzer.

## INTRODUCTION

Korean is an agglutinative language in which words consist of several morphemes, and some verb forms change when conjugated with functional suffixes. A Korean morphological analyzer (KMA) is designed to analyze a word and identify functional morphemes, which can specify the syntactic role of words in a sentence. Although end-to-end approaches are widely used in deep-learning models, some applications such as syntactic parsers require a KMA as a preprocessor to separate functional morphemes before parsing.

In many cases, the productive inflectional system in Korean causes deletion and contraction between a stem and the following morphemes when creating a Korean word. Therefore, a KMA should identify the base form of a morpheme by recovering deleted morphemes and decomposing contracted morphemes (*Han & Palmer, 2004*). Therefore, a KMA output sequence differs from a raw input sequence in both length and surface form. Figure 1 shows an example of KMA input and output. *INPUT* is a sequence of words separated by white spaces, and *OUTPUT* is a morphologically analyzed result. The

Corresponding author
Kong Joo Lee, kjoolee@cnu.ac.kr

**INPUT:**  까만        갤노트를          잃어버렸다
           *kka-man*   *kayl-no-thu-lul*   *ilh-e-pe-lyess-ta*
           black       galnote-*OBJ*        lost

( I ) lost a black galaxy note

**OUTPUT:**  까맣/VA   ㄴ/ETM   *<SP>*   갤노트/NNP   를/JKO   *<SP>*   잃어버리/VV   었/EP   다/EF
            black    *MOD*              galnote      *OBJ*              lose        *PAST*   *DEC*

**Figure 1** **The output of a morphological analyzer for the example input sentence "I lost a black galaxy note".** (The Yale Romanized system is used to transcribe Korean sentences. 'VA' is an adjective, and 'VV' is a verb. 'ETM' is an adnominal ending that attaches to the end of a verb or an adjective. 'NNP' is a proper noun, and 'JKO' is a particle that attaches to the end of nouns indicating an objective case. 'EP' is a verbal ending in the past tense, and 'EF' is a verbal ending to make a sentence declarative.).

output is a sequence of a single morpheme and the part-of-speech (POS) that follows the morpheme. The symbol "<SP>" indicates a word boundary. In this example, the input consists of three words, while the output consists of three words or seven morphemes, where some morphemes indicate grammatical relationships in a sentence.

Because the input and output lengths are different, Korean morphological analysis can be defined as an encoder–decoder problem. A raw input sequence is encoded and then decoded into a morphologically analyzed sequence. An encoder–decoder problem can be easily implemented by adopting two recurrent neural networks. Recent research in deep learning has proposed a new architecture, Transformer (*Vaswani et al., 2017*), for encoder–decoder problems that can increase the parallelism of learning processes by eliminating recurrent connections. Transformer calculates self-attention scores that can cross-reference between every input. Therefore, we adopt the Transformer architecture to implement a KMA in this work. To train a KMA based on Transformer from scratch, we need a considerable parallel corpus that includes raw input sentences and their analyzed results.

Since the introduction of pretrained language representation models such as Bidirectional Encoder Representations from Transformers(BERT) (*Devlin et al., 2019*), most natural language processing (NLP) applications have been developed based on pretrained models. Pretrained models provide context-dependent embeddings of an input sequence and reduce both the chronic problem of a lack of training data and the time required for training.

In this work, we utilize two types of BERT models to initialize Transformer, the backbone of a KMA. One is pretrained with Korean raw sentences and the other with morphologically analyzed sentences consisting of morphemes and POS tags. For the sake of clarity, we name the former "word-based BERT" (*w*BERT) and the latter "morpheme-based BERT" (*m*BERT); *m*BERT can encode a morphologically analyzed sequence into embedding vectors in the same way *w*BERT can encode a raw sentence. We initialized the Transformer encoder with *w*BERT and the Transformer decoder with *m*BERT.

While it is reasonable to initialize a Transformer encoder with BERT, it may seem unusual to initialize a decoder with BERT. We do not have decoder-based models like

GPT (*Radford et al., 2018*) pretrained with Korean morphologically analyzed data; instead, only *m*BERT is pretrained with Korean morphologically analyzed data. Recently, *Chi et al. (2020)* demonstrated that initializing an encoder and decoder with XLM (*Lample & Conneau, 2019*) produced better results than initializing them with random values. XLM is a pretrained model for cross-lingual tasks and is implemented based on a Transformer encoder.

Therefore, both an encoder and decoder of a KMA can be expected to benefit from initializing parameters with pretrained models. When Transformer is initialized with pretrained models, implementing a KMA based on the Transformer is a fine-tuning process that can be done with a relatively small corpus. In addition, employing the fine-tuning process is easier and faster than building a KMA from scratch.

Pretrained models such as BERT generally have 12–24 layers, which is deeper than conventional models. A few studies (*Clark et al., 2019*; *Jawahar, Sagot & Seddah, 2019*) have examined which layers of pretrained models are best suited for which tasks. It has been established that tasks dealing with words and surface forms of a sentence generally perform well on the lower layers rather than on the top layer. In this work, we also investigate the number of layers in a Transformer architecture that obtains the best accuracy for Korean morphological analysis.

Our contributions to achieving high-performance Korean morphological analysis are the following:

1. Because we leverage pretrained Korean language representation models to initialize the encoder and decoder of a morphological analyzer, we can train the morphological analyzer faster and with less training data.
2. We find the most appropriate number of layers in the BERT models for a KMA rather than using all layers in the models.

In the following section, we first explore related studies, and then we present the main architecture of a Korean morphological analyzer in 'A Korean Morphological Analyzer'. Experimental results are described in 'Experiments', followed by the conclusion in 'Conclusion'.

## SURVEY OF KOREAN MORPHOLOGICAL ANALYZERS

Traditional Korean morphological analysis consists of two pipeline stages: the first step is to separate morphemes from a word and convert them into their stems, and the second step is to assign them POS tags. Recently, a deep learning-based end-to-end approach has been applied to many applications, including KMAs. A sequence-to-sequence architecture based on recurrent neural networks is most often used to implement a KMA in full end-to-end style. Using this architecture, morphological analyzers can be easily implemented without complicated feature engineering or manually built lexicons. Conventional morphological analysis models suffer from the out-of-vocabulary (OOV) problem. To mitigate this problem, the following models adopted syllable-based sequence-to-sequence architecture.

*Li, Lee & Lee (2017)* adopted gated recurrent unit networks to implement a KMA with a syllable-based sequence-to-sequence architecture. In addition, an attention mechanism

(*Luong, Pham & Manning, 2015*) has been introduced to calculate the information needed by a decoder to ensure the model performs well.

*Jung, Lee & Hwang (2018)* also used syllable-level input and output for a KMA to alleviate the problem of an unseen word. Even with syllable-level input and output, the model tends not to generate characters that rarely occur in a training corpus. Therefore, they supplemented the model with a copy mechanism (*Gu et al., 2016*) that copies rare characters to output sequences. A copy mechanism assigns higher probabilities to rare or OOV words to perform better sequence generation during decoding phases. They reported that the accuracy of the KMA improved from 95.92% to 97.08% when adopting input feeding and the copy mechanism.

*Choe, hoon Lee & goo Lee (2020)* proposed a KMA specially designed to analyze Internet text data with several spacing errors and OOV inputs. To handle newly coined words, acronyms, and abbreviations often used in online discourse, they used syllable-based embeddings, syllable bigrams, and graphemes as input features. The model performed better when the dataset was collected from the Internet.

Since the introduction of BERT (*Devlin et al., 2019*) in the field of NLP research, pre-training-then-fine-tuning approaches have become prevalent in most NLP applications. In addition, notable improvements have been reported in several studies that have adopted the pre-training-then-fine-tuning framework. However, due to the distinct characteristics of Korean complex morphology systems, previous KMA studies have not adopted large-scale pretrained models such as BERT. *Li, Lee & Lee (2017)*, *Jung, Lee & Hwang (2018)*, and *Choe, hoon Lee & goo Lee (2020)* initialized word embeddings with random values and then trained them through a supervised learning step. *Park, Lee & Kim (2019)* partially adopted only character embeddings from BERT because BERT's sub-word units and Korean morphemes differ from each other. In this work, we implement a KMA using Transformer. In addition, our work is the first attempt to initialize both an encoder and decoder of Transformer with two different types of Korean BERT, which are $w$BERT and $m$BERT.

## A KOREAN MORPHOLOGICAL ANALYZER

Figure 2 illustrates the basic architecture of a KMA. Our model is implemented based on Transformer, which consists of an encoder and a decoder. The encoder, the inputs of which are raw sentences, is initialized with $w$BERT and the decoder, the outputs of which are morphologically analyzed sentences, is initialized with $m$BERT, as shown in Fig. 3.

A training dataset consists of pairs of raw input sentences with their morphological analyzed sequences. First, both sequences are tokenized into the multiple sub-word tokens (WordPiece, *Wu et al., 2016*) used by $w$BERT and $m$BERT. A set of WordPiece that both $w$BERT and $m$BERT use does not include a word-separator token to indicate word boundaries. However, when the decoder generates a sequence of morpheme tokens, the word separator token "<SP >" must be specified between morpheme tokens to recover word-level results, as shown in the output of Fig. 1.

Therefore, we adopt a multi-task learning approach to generate morphological analysis results while also inserting word-separators between the results. On the final layer of the

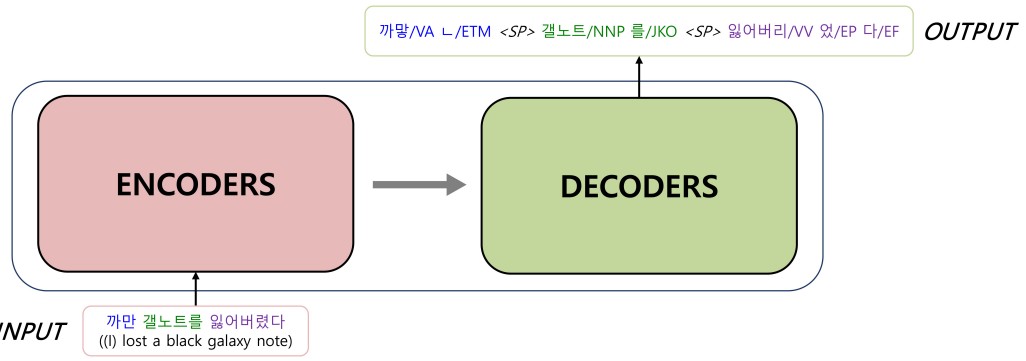

**Figure 2** Transformer: the basis architecture of a Korean morphological analyzer.

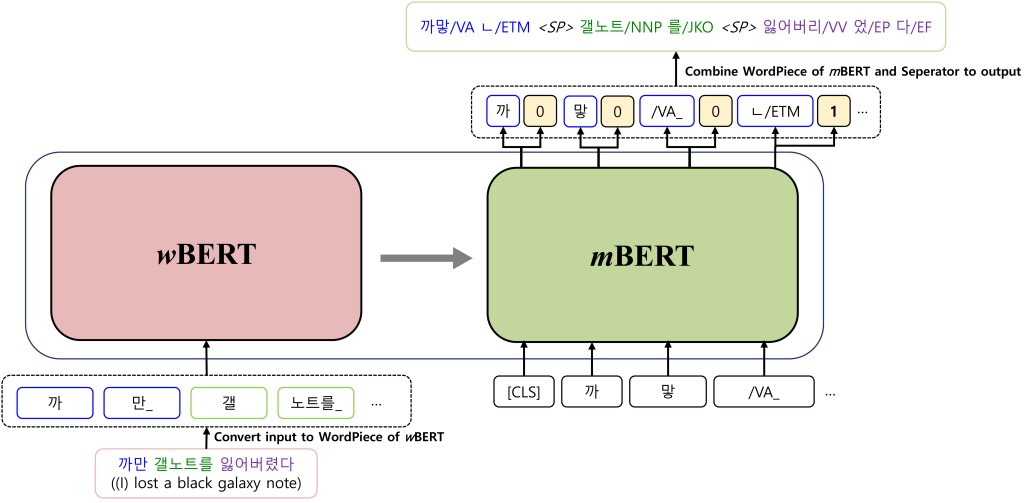

**Figure 3** Initialization of the Korean morphological analyzer with *w*BERT and *m*BERT.

decoder, there is an additional binary classifier that can discern whether a word-separator is needed for each output token. The final layer of the decoder produces two types of output, as shown in Fig. 3, which are combined to generate a final morphological analyzed sequence.

Given a raw input sentence, the tokenizer of *w*BERT splits words into multiple sub-word tokens. A token input to *w*BERT consists of a vector summation of a token embedding ($w$), a positional embedding ($p$), and a segment embedding ($E_A$), as shown in Eq. (1).

$$x_i = w_i + p_i + E_A \tag{1}$$

where $i$ is a token index of a sentence. The input of the encoder is denoted by $X = \{x_1, x_2, .., x_n\}$. The encoder of Transformer encodes $X$ into a sequence of contextualized embedding vectors $Z = \{z_1, z_2, \ldots, z_n\}$. Let us denote a sequence of hidden states in the decoder by $H = \{h_1, h_2, \ldots, h_m\}$. As this work adopts a multi-task approach, the decoder

simultaneously produces two kinds of output. The output of morpheme tokens is denoted by $Y = \{y_1, y_2, \ldots, y_m\}$ and the output of word separators is denoted by $S = \{s_1, s_2, \ldots, s_m\}$, where $s_i \in \{0, 1\}$. At time $t$ of the decoding phase, an output morpheme and a word-separator are determined by Eqs. (2) and (3), respectively.

$$P_G(y_t|X) = P(y_t|Z, Y_{<}t) = softmax(W_G^T h_t) \tag{2}$$

$$P_S(s_t|X) = P(s_t|Z, Y_{<}t) = softmax(W_S^T h_t) \tag{3}$$

where $W_G$ and $W_S$ are learnable parameters for generating both outputs. The objective function $L$ of the model is in Eq. (4)

$$
\begin{aligned}
L_{Gen} &= -\sum_{t=1}^{m} \log P_G(y_t|X) \\
L_{Sep} &= -\sum_{t=1}^{m} (\log P_S(s_t|X) - \log(1 - P_S(s_t|X))) \\
L &= L_{Gen} + L_{Sep}
\end{aligned}
\tag{4}
$$

Figure 4 provides an example to clearly understand the model. The raw input sentence has three words, which are split into eight WordPiece tokens (shown in *INPUT(X)*) for the input of the encoder. The symbol '_' in *INPUT(X)* indicates a word boundary. The output of the KMA is *OUTPUT(Y)*. The symbol '_' in *OUTPUT(Y)* is not a word boundary but a morpheme boundary. The KMA has another output, *OUTPUT(S)*, that predicts whether it is a word boundary or not. A '1' in *OUTPUT(S)* indicates a word boundary, where we insert "<SP >" when generating the final result.

## EXPERIMENTS

### Datasets and experimental setup

We used 90,000 sentences for training, 1,000 sentences for validation, and 10,000 sentences for evaluation in this work. They were all collected from the POS-tagged corpus published by the 21st Century Sejong Project (*Kim, 2006*). The sentence lengths were all less than 100 words and 46 POS labels were used in the Sejong corpus.

Table 1 describes the average number and the maximum number of WordPiece tokens in the sentences. We adopted both *w*BERT and *m*BERT, released by the Electronics and Telecommunications Research Institute (https://aiopen.etri.re.kr). They were pretrained with approximately 23 GB of data from newspapers and Wikipedia.

Table 2 shows the hyperparameters of the encoder and decoder, and Table 3 shows the hyperparameters for training the model.

The following experiments were designed to find the combination of the numbers of encoder and decoder layers that achieved the best KMA performance. First, we initialized the Transformer encoder and decoder with *w*BERT and *m*BERT, respectively, including embeddings of the WordPiece tokens. The cross attentions between the Transformer encoder and decoder were initialized with random values. Because the encoder and

| INPUT: | 까만 | 갤노트를 | 잃어버렸다 |
|---|---|---|---|
| | black | galnote-*OBJ* | lost |
| | *( I ) lost a black galaxy note* | | |
| INPUT(X): | 까 만_ | 갤 노트를_ | 잃 어 버 렸다_ |

| | | | | | | | | | | | |
|---|---|---|---|---|---|---|---|---|---|---|---|
| OUTPUT(Y): | 까 | 맣 | /VA_ | ㄴ/ETM_ | 갤 | 노트/NNP_ | 를/JKO_ | 잃 | 어 | 버리/VV_ | 었/EP_ | 다/EF_ |
| OUTPUT(S): | 0 | 0 | 0 | 1 | 0 | 0 | 1 | 0 | 0 | 0 | 0 | 0 |

| | | | | | | | |
|---|---|---|---|---|---|---|---|
| Final Result: | 까맣/VA | ㄴ/ETM | <SP> | 갤노트/NNP | 를/JKO | <SP> | 잃어버리/VV | 었/EP | 다/EF |
| | black | *MOD* | | galnote | *OBJ* | | lose | *PAST* | *DEC* |

**Figure 4** The input and the output of the Korean morphological analyzer for "I lost a black galaxy note".

**Table 1** The statistics of the dataset.

| Corpus | | Number of sentences | Average number of tokens | Maximum number of tokens |
|---|---|---|---|---|
| **Train** | Input | 90,000 | 34.80 | 220 |
| | Output | | 39.22 | 185 |
| **Validation** | Input | 1,000 | 40.61 | 120 |
| | Output | | 45.16 | 126 |
| **Evaluation** | Input | 10,000 | 34.06 | 126 |
| | Output | | 38.55 | 132 |

**Table 2** The hyperparameters of the *w*BERT and *m*BERT.

| Hyperparameters | Encoder (*w*BERT) | Decoder (*m*BERT) |
|---|---|---|
| Number of layers | 1–12 | 1–12 |
| Hidden dimension | 768 | 768 |
| Intermediate dimension | 3,072 | 3,072 |
| Number of attention heads | 12 | 12 |
| Activation function | Gelu | Gelu |
| Dropout | 0.1 | 0.1 |
| Maximum of input length | 512 | 512 |
| Vocabulary size | 30,797 | 30,349 |

decoder each had 12 layers, we compared the KMA performances by performing $12 \times 12$ combinations of encoder and decoder layers. When we adopted fewer than 12 layers of the encoder and decoder, we used the parameters of the corresponding layers from the bottom of *w*BERT and *m*BERT. The remaining parameters, such as $W_G$ and $W_S$ in Eqs. (2) and (3), were randomly initialized.

**Table 3** The hyperparameters for training the model.

| Hyperparameters | Value |
|---|---|
| Batch size | 64 |
| Optimizer | Adam |
| Learning rate(encoder, decoder) | 5e−3, 1e−3 |
| Beta1, Beta2 | 0.99, 0.998 |
| Maximum number of training steps | 100,000 |

## Results and evaluation

The BERT base model has 12 layers. *Jawahar, Sagot & Seddah (2019)* reported that tasks dealing with surface information performed best in the third and fourth layers of BERT, while tasks related to semantic information performed best in the seventh layer and above.

First, we wanted to find the optimal number of encoder and decoder layers to achieve the best morphological analysis performance while reducing the number of parameters to be estimated.

In the first experiment, the encoder and decoder were initialized with $w$BERT and $m$BERT, respectively. Then we compared the KMA performance according to the number of layers. The overall results of $12 \times 12$ combinations of encoder and decoder layers are shown in Fig. 5. The accuracy improved as the number of encoder layers increased, while it remained nearly the same when the number of decoder layers increased. We claim that KMA achieves the best performance with 12 encoder layers. In the second experiment, we examined the effect of the number of decoder layers on the KMA performance. We took a closer look at the results of Fig. 5 to examine the effect of the number of decoder layers on the KMA performance.

Based on the results of Fig. 5 the first experiment, we set the number of encoder layers at 12 and initialized it with $w$BERT. Then, we initialized the decoder with $m$BERT and compared the KMA performance while varying the number of decoder layers from 3 to 12. Table 4 shows the KMA accuracy according to the number of decoder layers. To get definitive results, we obtained the accuracies by averaging three trials for each row in Table 4. Although there seems to be little difference in KMA accuracy according to the number of decoder layers, the KMA performed best with four decoder layers. Surprisingly, the number of parameters in the decoder can be reduced without deteriorating the KMA performance.

Through the above experiment these two experiments, we arrived at the preliminary conclusion that KMA performs best when the decoder has only four layers and the encoder has the full number of layers.

Table 5 summarizes the experimental results to evaluate the impact of initializing $w$BERT and $m$BERT. In all subsequent experiments, the encoder had 12 layers and the decoder had four. The KMA significantly improved when the encoder was initialized with $w$BERT, increasing the F1 score by as much as 2.93. There was a slight performance improvement when the decoder was initialized with $m$BERT. The KMA with the four-layer decoder outperformed the KMA with the full-layer decoder. In addition, we discovered

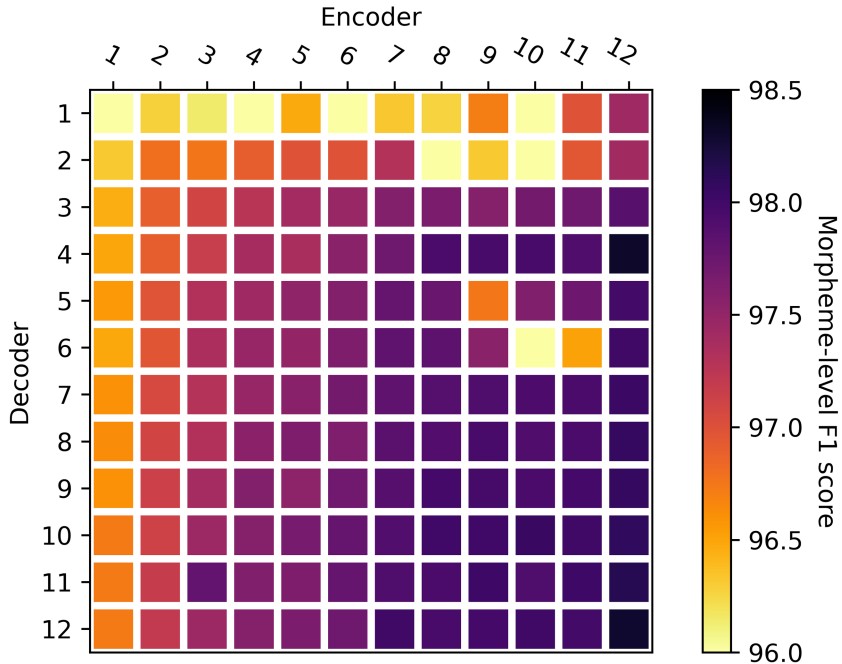

**Figure 5** **Comparison of performances according to the number of encoder and decoder layers.** The *X*-axis shows the number of encoder layers, and the *Y*-axis shows the number of decoder layers.

**Table 4** **Comparison of accuracies according to the number of decoder layers.**

| Number of layers in the Decoder | Morpheme-level F1 score | Separator accuracy | Sentence-level accuracy | Number of parameters |
|---|---|---|---|---|
| 3 | 97.86 | 99.64 | 59.48 | 184M |
| 4 | 98.31 | 99.78 | 66.45 | 193M |
| 5 | 97.99 | 99.67 | 60.88 | 203M |
| 6 | 98.01 | 99.70 | 61.33 | 212M |
| 7 | 98.04 | 99.71 | 61.53 | 222M |
| 8 | 98.07 | 99.72 | 62.22 | 231M |
| 9 | 98.07 | 99.72 | 62.07 | 241M |
| 10 | 98.09 | 99.72 | 62.46 | 250M |
| 11 | 98.14 | 99.74 | 63.16 | 260M |
| 12 | 98.29 | 99.73 | 64.42 | 269M |

that the word separator could easily exceed 99% accuracy by adopting multi-task learning. We obtained approximately 93.58% accuracy in word separation when adopting an independent word separator.

Table 6 presents the robustness of the KMA to the lengths of the input sentences. The KMA with 12 encoder layers and four decoder layers outperformed the other combinations of models in evaluating longer inputs.

To show the effect of initializing parameters with pretrained models more clearly, we performed the following experiments. We measured changes in the accuracy of the models

**Table 5** The comparison of accuracies according to the number of layers and initialization in the encoder and the decoder.

| Number of layers & Initialization in the encoder and decoder | Morpheme-level F1 score | Separator accuracy | Sentence-level accuracy |
|---|---|---|---|
| Encoder-12-random Decoder-4-random | 95.32 | 98.84 | 40.49 |
| Encoder-12-*w*BERT Decoder-4-random | 98.25 | 99.79 | 63.91 |
| Encoder-12-*w*BERT Decoder-4-*m*BERT | 98.31 | 99.78 | 66.45 |
| Encoder-12-*w*BERT Decoder-12-*m*BERT | 98.29 | 99.73 | 64.42 |

**Table 6** Comparison of accuracies according to the input length and number of layers and initialization in the encoder and the decoder.

| Encoder, Decoder Initialization | Input sentence length | | |
|---|---|---|---|
| | 1–34 tokens | 35–100 tokens | >100 tokens |
| Encoder-12-random Decoder-4-random | 95.80 | 95.45 | 86.66 |
| Encoder-12-*w*BERT Decoder-4-random | 98.25 | 98.26 | 97.35 |
| Encoder-12-*w*BERT Decoder-4-*m*BERT | 98.35 | 98.31 | 97.37 |
| Encoder-12-*w*BERT Decoder-12-*m*BERT | 98.14 | 98.11 | 96.28 |

**Table 7** Comparison of F1 scores according to the size of the training corpus.

| Encoder, Decoder Initialization | Training dataset size | | | | |
|---|---|---|---|---|---|
| | 10% | 30% | 50% | 70% | 100% |
| Encoder-12-random Decoder-4-random | 26.57 | 85.66 | 92.25 | 94.38 | 95.32 |
| Encoder-12-*w*BERT Decoder-4-random | 93.40 | 96.91 | 97.57 | 97.80 | 98.25 |
| Encoder-12-*w*BERT Decoder-4-*m*BERT | 95.23 | 97.37 | 97.87 | 98.04 | 98.31 |
| Encoder-12-*w*BERT Decoder-12-*m*BERT | 94.84 | 97.17 | 97.71 | 97.86 | 98.29 |

as the size of the training dataset decreased from 100% to 10%. The results are shown in Table 7.

Initializing the model with *w*BERT and *m*BERT degraded the accuracy by less than 0.5%, even when only half of the training dataset was used to train the model. For the model initialized with *w*BERT alone, the accuracy deteriorated by more than 0.6%, while the accuracy of the model initialized with random values decreased by more than 3%. The

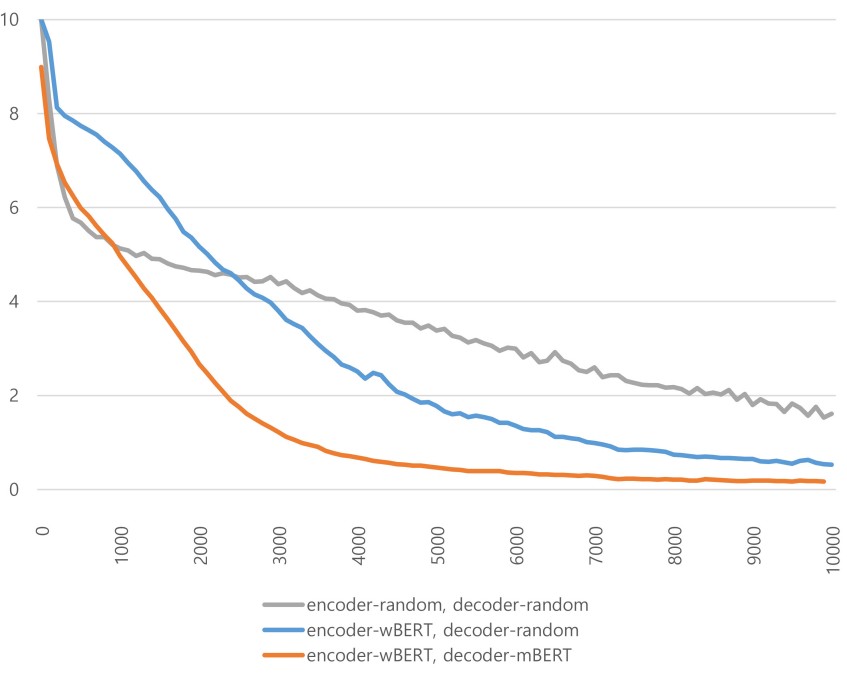

**Figure 6  Comparison of performances according to the number of encoder and decoder layers.** The $X$-axis shows the number of encoder layers, and the $Y$-axis shows the number of decoder layers.

most surprising result was that the model initialized with $w$BERT and $m$BERT achieved 95.23% accuracy when trained with only 10% of the training dataset. This result was approximately the same as when the model was trained with the complete training dataset after being initialized with random values.

As the size of training dataset becomes smaller, the performance of the model in the last row decreases faster than the model with 12 and 4 layers in the encoder and decoder initialized with $w$BERT and $m$BERT, respectively. The main reason is that the former has more parameters than the latter, and models with more parameters require more training data.

Figure 6 depicts the training loss curves of the models with different initialization values depending on the number of training steps. The $X$-axis shows the number of training steps, while the $Y$-axis shows the loss values of the training steps. At the beginning of the training phase, the losses of the model initialized with random values decreased very sharply. However, as the number of training steps increased, the loss of the model using $w$BERT and $m$BERT as parameter initializers dropped faster than the other models. This shows that the KMA can be trained robustly and efficiently when its parameters are initialized with the pretrained models $w$BERT and $m$BERT. Parameter initialization with pretrained models helps successfully build deep learning models.

Table 8 shows a comparison of the F1 scores with those of previous approaches that used sequence-to-sequence architectures. We reimplemented the models from the previous approaches and compared their results directly to those of our model in the same

**Table 8 Comparison of results of previous approaches to those of the proposed model.**

| Models | F1 score (re-implementation) |
| --- | --- |
| Sequence-to-sequence (syllable-basis) (*Li, Lee & Lee, 2017*) | 97.15 (96.58) |
| Sequence-to-sequence (syllable-basis) + input feeding + copy mechanism (*Jung, Lee & Hwang, 2018*) | 97.08 (96.67) |
| Sequence-to-sequence Syllable + grapheme + bigram embeddings (*Choe, hoon Lee & goo Lee, 2020*) | 97.93 (96.74) |
| Our model Transformer + *w*BERT + *m*BERT | **98.31** |

environments. The KMA proposed in this study demonstrates competitive end-to-end performance without any additional knowledge or mechanisms.

Table 9 in the appendix provides the accuracy comparisons between the multi-task separator and the independent separator.

## CONCLUSION

In this work, we suggested adopting the Transformer architecture to implement a KMA. Furthermore, we proposed using two Korean BERTs to initialize the parameters of the Transformer encoder and decoder. We introduced a multi-task learning approach to specify word boundaries in an output sequence of morpheme tokens. The KMA achieved its best performance when initialized with two types of Korean BERT. In addition, we observed that the accuracy of the KMA was highest when it had four layers in the decoder and 12 layers in the encoder. To conclude this work, we proved that appropriate parameter initialization can help ensure stable, fast training, and good performance of deep-learning models.

# APPENDIX

**Table 9  Comparison of accuracies according to the word separators. The multi-task separator was the same as that in Table 5.** The independent classifier was a binary classifier built on *m*BERT networks. The input of the independent classifier was the sequence of tokens without word boundary information, and the output was the sequence of prediction of whether each token was a word boundary.

| Word separators | Accuracy |
|---|---|
| Multi-task separator | 99.78 |
| Independent classifier | 93.58 |

## Funding

This results was supported by "Regional Innovation Strategy (RIS)" through the National Research Foundation of Korea (NRF) funded by the Ministry of Education (MOE)(2021RIS-004). The funders had no role in study design, data collection and analysis, decision to publish, or preparation of the manuscript.

## Grant Disclosures

The following grant information was disclosed by the authors:
The National Research Foundation of Korea (NRF) funded by the Ministry of Education (MOE): 2021RIS-004.

## Competing Interests

The authors declare there are no competing interests.

## Author Contributions

- Yong-Seok Choi conceived and designed the experiments, analyzed the data, prepared figures and/or tables, and approved the final draft.
- Yo-Han Park performed the experiments, analyzed the data, performed the computation work, prepared figures and/or tables, and approved the final draft.
- Kong Joo Lee conceived and designed the experiments, authored or reviewed drafts of the paper, and approved the final draft.

## Data Availability

The code is available at Github: https://github.com/yseokchoi/KMAwithBERTs.
The data is available at everyone's corpus: https://corpus.korean.go.kr/main.do.

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
