# Peer review of "Building a Korean morphological analyzer using two Korean BERT models"

_PeerJ Computer Science, doi:10.7717/peerj-cs.968_

## Round 0.1 · original submission · Major Revisions

The authors are advised to make Major Revisions to the manuscript, as per the given comments from the reviewers, and resubmit the paper.

·

Basic reporting

In this paper, a Korean morphological analyzer (KMA) is designed to output a sequence of a single morpheme and its Part-of-speech (POS) that follows the morpheme. As the lengths of the input and output are different, Korean morphological analysis can be defined as an encoder-decoder problem.In this work,the author adopt the Transformer architecture to implement a KMA.
The author's contributions are as follows:
1. In this work, the author utilized two kinds of BERT models to initialize Transformer. The wBERT is pre-trained with Korean raw sentences and the mBERT is pre-trained with morphologically analyzed sentences consisting of morphemes and POS tags.And the author initialized an encoder of Transformer with wBERT and a decoder of Transformer with mBERT.
2. The author found the optimal number of layers for the encoder and decoder to achieve the best performance of KMA.
3. The author summarized the experimental results that evaluate the impact of initializing wBERT and mBERT.
4. The author measured changes in accuracy in the models as the size of training dataset decreased from 100% to 10%.

Experimental design

Strengths and Weaknesses
1. In the experiment of finding the optimal number of layers for the decoder,the encoder is initialized with wBERT while the decoder is randomly initialized.But there seems to be no reason to do so, so I think the author should explain the specific reasons for it.
2. In the experiment of finding the optimal number of layers for the encoder, the author fixed the number of layers in the decoder at 4. So the conclusion of this experiment is based on the premise that decoder has 4 layers, and it doesn't mean that the optimal number of layers is that(maybe when the decoder has other layers).
3. In table 7, when wBERT and mBERT are used to initialize the model and the data set is 10%, the data in this cell should be 95.23% instead of 98.23%.
4. In table 8, the comparation of F1 values in different models is meaningless, because they used different test datasets.

Validity of the findings

Innovation
This paper is the first attempt to initialize both an encoder and decoder for Transformer with two kinds of Korean BERT.

·

Basic reporting

The structure of the overall article is very logical, but there are some problems in the details.
(1)In the relevant work, the analysis of the relevant research on Korean at this stage is not very clear, and there is no clear analysis of the research deficiencies at this stage and the innovation of this research. It is necessary to explain the enlightenment or reference of the past literature research to this research.
(2)The paper does not explain how many words the longest text and the shortest text of the experimental data are respectively. It is necessary to further explain whether the algorithm is suitable for long text or short text. According to the experimental data set in the paper, it seems that it is only suitable for the training of short text, so we need to focus on this aspect again.

Experimental design

(1)The specific process description of the experiment is a little simple, such as pretreatment, model experiment process, etc., which should be connected with the subsequent evaluation.
(2) In the introduction section, in the second paragraph, in order to verify the problem of "the output sequence of a KMA differs from a raw input sequence in both length and surface form.", I think that not all sentences are the same problem, please add which one kind of sentence has this problem.
(3) In the section of SURVEY ON KOREAN MORPHOLOGICAL ANALYZER, it only proves the rationality of the transform structure, but does not prove the rationality of "initialize both an encoder and decoder for Transformer with two kinds of Korean BERT.", please add relevant explanations.

Validity of the findings

(1) The main contribution of this article is "train the morphological analyzer faster and with less training data".In Table 7, the size of the data set is divided into 10%, 30%, 50%, 70%, 100%. The F1 score first drops and then rises, but the F1 value of the data set at 100% is significantly better than the F1 value of 10%, please Explain why the model proposed in this paper requires a smaller amount of data set.
(2)For the main contribution 2 "find the most appropriate number of layers in the BERT models for a Korean morphological analyzer"&" we observed that the accuracy of the Korean morphological analyzer is highest when it has four layers in the decoder and 12 layers in the encoder."Compare the experiment, whether the data set in other fields is applicable to the results of this article.

Additional comments

No comment.

Reviewer 3 ·

Basic reporting

some details are missing in this version.
the reviewer believes that the introduction is overlength.

Experimental design

some mismatched data in the experiment section.

Validity of the findings

the proposed model is worked in a certain dataset.
more experiments are required to validate the proposed model.

Additional comments

1 the author should clarify how the encoder and decoder are initialized with the pre-trained BERT model.
2 it is really necessary to define the classification between the word separator token and morpheme tokens as a multi-task learning? The authors should provide sufficient rationale.
3 more experiments should be conducted to validate the multi-task learning in this work.
4 In section experiment, the data (line 180) is mismatched with that of in Tab.7.
5 more details should be provided to improve the readability of this work, such as the initialization.

---

## Round 0.2 · Major Revisions

Based on reviewer's comments, changes are required for your manuscript.

·

Basic reporting

Compared with the previous version, the author has made some supplements and modifications to the experiment, but there are still some problems in the details.

(1)When exploring the influence of training set size on experimental results, there should be experiments with full-layers for both encoder and decoder, as in the previous experiments.


From my point of view,  I suggest a few modifications are still needed before publication.

Experimental design

(1)In the experiment of finding the optimal number of encoder and decoder layers , I think the two experiments made by the author can be combined into one, that is, the optimal combination of layers for encoder and decoder can be determined by the first experiment alone.

Validity of the findings

Compared with the previous version, the author has made some supplements and modifications to the experiment, but there are still some problems in the details.

Additional comments

(2)In table 6, the fourth column heading should be '> 100 tokens' not '< 100 tokens'.

---

## Round 0.3 · accepted · Accept

Based on the revisions made by the authors, the paper is accepted.